# Evaluating Distributional Distortion in Neural Language Modeling

**Benjamin LeBrun[1,2,†]  Alessandro Sordoni[3,*] & Timothy J. O'Donnell[1,2,4,*]**
[1]McGill University  [2]Mila – Quebec Artificial Intelligence Institute  [3]Microsoft Research
[4]Canada CIFAR AI Chair, Mila

## Abstract

A fundamental characteristic of natural language is the high rate at which speakers produce novel expressions. Because of this novelty, a heavy-tail of rare events accounts for a significant amount of the total probability mass of distributions in language (Baayen, 2001). Standard language modeling metrics such as perplexity quantify the performance of language models (LM) in aggregate. As a result, we have relatively little understanding of whether neural LMs accurately estimate the probability of sequences in this heavy-tail of rare events. To address this gap, we develop a controlled evaluation scheme which uses generative models trained on natural data as artificial languages from which we can exactly compute sequence probabilities. Training LMs on generations from these artificial languages, we compare the sequence-level probability estimates given by LMs to the true probabilities in the target language. Our experiments reveal that LSTM and Transformer language models (i) systematically underestimate the probability of sequences drawn from the target language, and (ii) do so more severely for less-probable sequences. Investigating where this probability mass went, (iii) we find that LMs tend to overestimate the probability of ill-formed (perturbed) sequences. In addition, we find that this underestimation behaviour (iv) is weakened, but not eliminated by greater amounts of training data, and (v) is exacerbated for target distributions with lower entropy.

## 1 Introduction

Natural language is fundamentally creative—speakers and listeners frequently produce and comprehend sentences which have never been produced before (Fodor, 1975; Fodor & Pylyshyn, 1988; Chomsky, 1975, 1955). As a side-effect of this property, distributions in natural language are characterized by a *heavy-tail* of individually improbable events which collectively account for a significant amount of the total probability mass of the distribution (Khmaladze, 1988; Baayen, 2001). Precisely approximating this large number of rare events is one of the foundational challenges for models of natural language (Good, 1953; Jelinek, 1980; Katz, 1987; Kneser & Ney, 1995; Wood et al., 2011; Goldwater et al., 2011). *Autoregressive neural language models* (Bengio et al., 2003; Mikolov et al., 2013; Radford et al., 2019) attempt to do so by decomposing the probability of an event (a sequence) into a series of conditional distributions, each parameterized by a shared neural network.

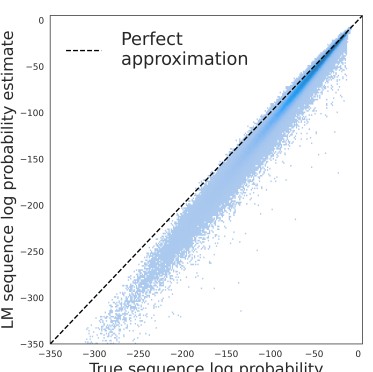

Figure 1: GPT2 sequence probability estimates plotted against the true sequence probabilities. Neural LMs underestimate the probability of sequences drawn from the language they are trained to model.

Recently, a growing body work has sought to understand how these language models (LM) fit the distribution of a language beyond standard measures such

[†]Corresponding author: benjamin.lebrun@mail.mcgill.ca
[*]Co-senior autorship

as *perplexity*. Meister & Cotterell (2021), for example, investigated the statistical tendencies of the distribution defined by neural LMs, whereas Kulikov et al. (2021) explored whether they adequately capture the modes of the distribution they attempt to model. At the same time, increased focus has been given to performance on rare or novel events in the data distribution, both for models of natural language (McCoy et al., 2021; Lent et al., 2021; Dudy & Bedrick, 2020; Oren et al., 2019) and neural models more generally (see, for example Sagawa et al., 2020; D'souza et al., 2021; Chen et al., 2021; Blevins & Zettlemoyer, 2020; Czarnowska et al., 2019; Horn & Perona, 2017; Ouyang et al., 2016; Bengio, 2015; Zhu et al., 2014). Neither of these branches of work, however, has explored instance-level LM performance on rare sequences in the distribution. As a result, we have relatively little understanding of how neural LMs approximate sequences in the heavy-tail characteristic of natural language.

In this work, we introduce a controlled methodology to explore how LMs estimate the probability of sequences in the heavy-tail of the distribution. Our instance-level evaluation scheme explicitly compares the target probability distribution of the language to the distribution defined by the LM. Since the true distribution of any natural language is in practice unknown, we use a Transformer LM trained on natural data as a generative model to define target *artificial languages* for which we can exactly compute sequence probabilities. Training LSTM and Transformer LMs on sequences sampled from these target artificial languages, we compare the sequence-level probability estimates given by neural LMs to the target probabilities in the language. By controlling the entropy of the generative model's conditional distributions, we create a set of artificial languages with varying distributional properties, and analyze how LM estimation behaviour is modulated by the properties of the target distribution.

Our experiments uncover the extent to which neural LMs provide a distorted fit of the language they are trained to model. We find that LSTM and Transformer LMs (i) systematically underestimate the probability of sequences drawn from the target language and (ii) do so more when such sequences are rare. Where did this underestimated probability mass go? We do not find that the underestimation is accompanied by overestimation in the head of distribution. Rather, we find that LMs tend to (iii) overestimate the probability of rare perturbed (ill-formed) sequences. Interpreted together, these findings indicate that on the one hand, neural LMs under-represent well-formed sequences in the tail of the language they attempt to model, and on the other hand, over-represent ill-formed sequences far away from high probability zones in sequence-space. In addition, we find that (iv) greater amounts of training data lessen underestimation but do not eliminate it and that (v) underestimation is exacerbated for target distributions with lower entropy.

## 2 BACKGROUND

We begin by briefly characterizing why distributions with a large number of rare events (LNRE) emerge in natural language, and why these events pose challenges for LMs. Furthermore, we motivate the need for instance-level evaluation when dealing with a large number of rare events.

**Productivity**   In the context of language production, a language user has the ability to produce, at any given point in their linguistic lifespan, an utterance which they have never produced before. This creativity is the result of the generative property of *productivity*, which states that on the basis of finite linguistic experience, a language user can produce and comprehend an unbounded number of grammatically acceptable utterances (Chomsky, 1975, 1955). Productive processes induce a distribution which places non-zero probability on unseen events at all practical sample sizes. Because of this property, many of the distributions in natural language—particularly the distribution over the sequences of a language—are characterized by a heavy-tail of rare events.

**LNRE Zone**   To make explicit the connection between productivity and a heavy-tail of rare events, let $\mathcal{P}_N$ denote the probability of sampling a novel (previously unseen) event from some distribution after having sampled $N$ events. Then productivity as described above states that $\mathcal{P}_N > 0$ for all sample sizes $N$ that occur in practice. The range of sample sizes $N$ for which it is the case that $\mathcal{P}_N > 0$ is known as the *LNRE zone* (Khmaladze, 1988; Baayen, 2001). The LNRE zone for natural language appears to be very large, and it seems likely that $\mathcal{P}_N$ will remain greater than 0 for samples of natural language many orders of magnitude larger than all the data currently available

for training LMs.[1] In the LNRE zone, it is difficult to obtain accurate estimates of the probability of events using straightforward maximum likelihood estimation (MLE). Accounting for this enormous amount of novelty is thus a central challenge in language modeling.

**Language modeling** A model $M$ of the language $L$ attempts to define a distribution $p_M$ which closely resembles the true distribution of the language $p_L$. In a locally-normalized autoregressive model, this distribution is defined by assigning probabilities to variable length sequences $\boldsymbol{x}$ via a chain-rule decomposition:

$$p(\boldsymbol{x}) = \prod_{i=1}^{|\boldsymbol{x}|} p(x_i \mid \boldsymbol{x}_{1:i-1}) = \prod_{i=1}^{|\boldsymbol{x}|} \frac{\exp \rho(\boldsymbol{x}_{1:i-1}, x_i)}{\sum_{x \in \Sigma} \exp \rho(\boldsymbol{x}_{1:i-1}, x)} \tag{1}$$

where $\Sigma$ is the vocabulary, and $\rho(\boldsymbol{x}_{1:i-1}, x_i; \theta)$ is the non-negative score of token $x_i$ given the sequence prefix $\boldsymbol{x}_{1:i-1}$, which is computed by a neural network with parameters $\theta$.

For $p_M$ to perfectly approximate $p_L$, we expect $p_M(\boldsymbol{x}) = p_L(\boldsymbol{x})$ for all $\boldsymbol{x} \in \Sigma^*$, where $\Sigma^*$ is the set of all strings of finite length (the Kleene closure of $\Sigma$). In the LNRE zone, $p_M$ is defined over a support containing a very large set of sequences which have never occurred in a training corpus (or equivalently, have all occurred with frequency 0), and which take on a very wide array of differing probabilities. For example, while the sequences $\boldsymbol{x}_1, \boldsymbol{x}_2 \in \Sigma^*$ have likely never occurred in any sample of English, most would agree that $\boldsymbol{x}_1$ is far more probable than $\boldsymbol{x}_2$:

$\boldsymbol{x}_1$: *The East pond in Parc Lafontaine was filled to the brim with Diet Coke.*

$\boldsymbol{x}_2$: *Certain nak indicate liberationing among theorter codity voters vandalized.*

**LM Evaluation** For a perfect LM of English, we would expect the estimated probabilities of the sequences $\boldsymbol{x}_1$ and $\boldsymbol{x}_2$ to match their probabilities under the true distribution $p_{\text{English}}$. However, since $p_{\text{English}}$ and it's underlying generative process are unknown, it is not possible to explicitly evaluate how closely instance-level probability estimates align. As a proxy, the mean perplexity of the model on a holdout set of sequences $\mathcal{D}$ is typically used, which measures whether the model, on average, assigns high likelihood to unseen instances. This measure does not, however, tell us whether instance-level estimates align with their true counterparts, nor is it necessarily indicative of performance on rare, idiosyncratic events in $\mathcal{D}$. In this way, the lack of access to the ground-truth distribution severely complicates LM evaluation on the heavy-tail of rare sequences in language. The following section introduces a methodology to overcome these limitations.

## 3 LANGUAGE MODEL EVALUATION IN THE LNRE ZONE

| Component | Notation | Description |
|---|---|---|
| Generative model | $L$ | A LM trained on natural instance-level data. |
| Artificial language | $p_L$ | The distribution over sequences induced by a sampling scheme from $L$. |
| Language model | $p_M$ | The distribution of a LM trained on sequences sampled from $p_L$. |
| Target probabilities | $p_L(\boldsymbol{x})$ | The probability assigned by $p_L$ to the sequence $\boldsymbol{x}$. |
| Model probabilities | $p_M(\boldsymbol{x})$ | The probability assigned by $p_M$ to the sequence $\boldsymbol{x}$. |

Table 1: Components of our instance-level evaluation scheme. Training $p_M$ on samples from $p_L$, we compare $p_M(\boldsymbol{x})$ to $p_L(\boldsymbol{x})$ for $x \in \Sigma^*$.

We propose evaluating language model performance on the heavy-tail of rare events via a known probability distribution over sequences. Specifically, we train a Transformer LM on sequences sampled from a corpus of natural language to define a generative model $L$. The distribution over sequences induced by a sampling scheme from $L$, denoted $p_L$, is then our *artificial language*. We expect a model $M$ of this artificial language to assign probabilities $p_M(\boldsymbol{x})$ to sequences $\boldsymbol{x}$ which match

---

[1]For an empirical validation of this claim on a sample of practical size from the OpenWebText corpus, see the Appendix.

the *target probabilities* $p_L(\boldsymbol{x})$ of $\boldsymbol{x}$ under $p_L$. To characterize neural LM behaviour on rare events, we train Transformer and LSTM LMs on data sampled from $p_L$, and compare the instance-level probability estimates given by $p_M$ to target probabilities under $p_L$. We summarize the components of this methodology in Table 1, and overview it in greater detail in the following section.

### 3.1    ARTIFICAL LANGUAGES AS TARGET DISTRIBUTIONS

| $\boldsymbol{x}$ | $\log p_L(\boldsymbol{x})$ |
|---|---|
| "we're very excited to have the opportunity to help them," he says. | $-29.3811$ |
| so what's going to happen? | $-17.4128$ |
| to me, the fisheries are in the midst of a global financial crisis. | $-41.4835$ |

Table 2: Sample of sequences drawn from our artificial language.

To define a generative model $L$, we train a randomly-initialized GPT2-medium on 1.5M sentences sampled from the OpenWebText corpus (Gokaslan & Cohen, 2019).[2] We set the maximum sequence length to be 128 tokens. We additionally train a byte-pair-encoding (BPE) tokenizer on this data set with a standard GPT2 vocabulary size of 57,256 tokens. For simplicity, this tokenizer is used for all models.

Using this generative model $L$, we define the target distribution over sequences—the artificial language—as the distribution induced by an ancestral sampling scheme from $L$. Thus, we draw instances $\boldsymbol{x} = (x_1, \ldots, x_{|\boldsymbol{x}|})$ from our language $p_L$ by recursively sampling from the conditional distribution over tokens at each time step: $x_t \sim p_L(\cdot \mid x_{<t})$ where $x_1 = \text{BOS}$ and $x_t = \text{EOS}$. All experiments up until Section 4.5 are conducted on the distribution induced by ancestrally sampling from $L$ with softmax temperature $T = 0.85$.[3] In Section 4.5, we explore the effects of different values of $T$ when sampling from $p_L$. Table 2 shows three sequences sampled from this distribution.

### 3.2    SEQUENCE PROBABILITY ESTIMATION IN THE LNRE ZONE

Given an artificial language $p_L$, the task of $M$—the language model—is to define a distribution $p_M$ whose sequence-level probability estimates closely align with the sequence-level probabilities given by $p_L$. We refer to any deviation from this desiderata as *model estimation error*. To quantify the model estimation error for a sequence $\boldsymbol{x}$, we take the difference between the sequence's log probability under $M$ and its true log probability under $L$:

$$\text{error}(\boldsymbol{x}) = \log p_M(\boldsymbol{x}) - \log p_L(\boldsymbol{x}) \tag{2}$$

This quantity is the log probability ratio, which measures, in log-space, the number of times more or less likely the sequence $\boldsymbol{x}$ is under the language model $M$. Note that $\text{error}(\boldsymbol{x}) < 0$ indicates that $M$ underestimates the probability of $\boldsymbol{x}$, whereas $\text{error}(\boldsymbol{x}) > 0$ indicates that $M$ overestimates the probability of $\boldsymbol{x}$. In practice, we train $M$ on a set of sequences $\mathcal{D}_{\text{train}}$ sampled from $p_L$, and compute model estimation error on a separate set of sequences $\mathcal{D}_{\text{test}}$ sampled from $p_L$. In all cases, we compute the probability of a sequence $\boldsymbol{x}$ as its chain rule decomposition: $p(\boldsymbol{x}) = \prod_{i=1}^{|\boldsymbol{x}|} p(x_i \mid x_{<i})$ where $x_0 = \text{BOS}$ and $x_{|\boldsymbol{x}|} = \text{EOS}$. When computing the ground-truth sequence probabilities for $p_L$, we take into account any softmax tempering.

### 3.3    NEURAL LANGUAGE MODELS

We study the estimation performance of two neural LM architectures: the Transformer (Vaswani et al., 2017) and the LSTM (Melis et al., 2020). When training either architecture, we halve the learning rate if validation loss increases at the end of an epoch. For all model sizes, we use a batch size of 128 sequences. Models with the lowest cross-entropy loss on a withheld validation set are used in experiments unless otherwise mentioned.

---

[2] All Transformer implementations were obtained from Huggingface, and training was done on two or four RTX-8000 GPUs (depending on model size) with mixed floating point precision.

[3] We temper in an effort to define a ground-truth distribution whose entropy more closely resembles that of a natural language. Note that our findings hold for untempered ($T = 1.00$) ground-truth distributions as well (see A.1.1 and A.1.2).

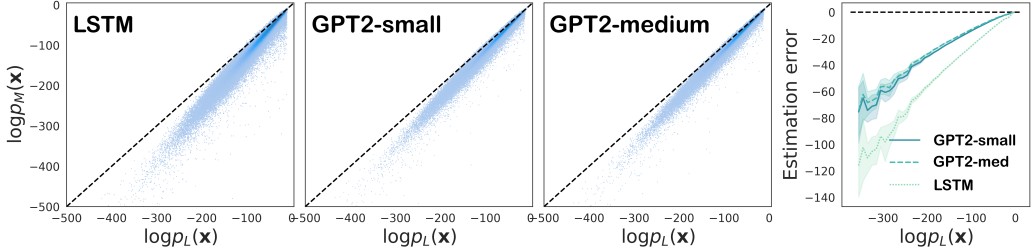

Figure 2: Test sequence probability estimates given by neural LMs. *Three left-most figures*: The joint histograms of sequence probability estimates. The dotted line denotes the cases in which the model's estimates perfectly align with the target probability; shading to the right of this line denotes underestimation. *Right-most figure*: Mean sequence estimation error by target sequence probability.

We use the Huggingface (Wolf et al., 2020) implementations of GPT2-small, GPT2-medium and GPT2-large (Radford et al., 2019) as representative Transformer LMs. We use Adam Optimization with $\epsilon = 1e^{-8}$ and learning rates $\alpha = 5e^{-5}$, $\alpha = 4e^{-5}$ and $\alpha = 3e^{-5}$ for GPT2-small, -medium and -large respectively. Since these models are in the same model class as our artificial target language $p_L$, we expect the task of recovering the ground-truth distribution to be relatively easy compared to the true problem faced in modeling natural language, where both the distribution and the underlying generative process are unknown. For the LSTM (Hochreiter & Schmidhuber, 1997), we follow the implementation of the baseline LM described in (Melis et al., 2020). We use 2 layers and adjust the hidden state and embedding dimension (2048 and 1024, respectively) to be such that the total number of parameters is approximately equal to GPT2-small (110M).

## 4 RESULTS

### 4.1 ESTIMATION ERROR WITH FIXED DATA

We begin by exploring model estimation error on a fixed training set $\mathcal{D}_{\text{train}}$ of 1M sequences sampled from $p_L$. We first train LSTM and GPT models on $\mathcal{D}_{\text{train}}$, early-stopping as described above. Following training, we sample a test set $\mathcal{D}_{\text{test}}$ of 500,000 sequences from $p_L$, and score each sequence under both the model distribution $p_M$ and the true language distribution $p_L$. From this, we obtain a set of probability estimates: $\mathcal{S}_{\text{test}} = \{\langle p_L(\boldsymbol{x}), p_M(\boldsymbol{x}) \rangle \mid \boldsymbol{x} \in \mathcal{D}_{\text{test}}\}$. If the model $M$ perfectly models the language $L$, then for each $\langle p_L(\boldsymbol{x}), p_M(\boldsymbol{x}) \rangle \in \mathcal{S}_{\text{test}}$ we would expect $p_L(\boldsymbol{x}) = p_M(\boldsymbol{x})$. Figures 2(A) and 2(B) visualize this relationship with the $x$- and $y$-axes denoting the true and model estimated sequence probabilities respectively, and a dashed line representing equality. To compare probability estimates, we represent the set $\mathcal{S}_{\text{test}}$ in the form of a joint histogram over this coordinate space. Histogram bins are shaded based on the number of tuples which lie in the coordinate range they define. Importantly, any deviation of this histogram from the identity line indicates that the model distorts the shape of the distribution of the language.

Figures 2(A) and 2(B) provide evidence for distributional distortion in the form of underestimation. The majority of probability tuples in $\mathcal{S}_{\text{test}}$ lie to the right of the identity line, indicating that LSTM and GPT2 models consistently underestimate the probability of sequences sampled from $p_L$. Furthermore, the distance between the identity line and the probability tuples grows non-linearly as function of the true sequence probability, indicating that underestimation error is more severe for rarer sequences in the language. We validate these observations in the right-most plot of Figure 2, which shows mean estimation error decreasing non-linearly as a function of the target sequences probability.[4] In addition, comparing underestimation behaviour across model size, we find that while GPT2-medium performs slightly better than GPT2-small, these improvements are typically within the range defined by the bootstrapped 95% confidence intervals. See A.1.3 for evidence indicating that this underestimation behaviour also occurs in pre-trained models fine-tuned on $\mathcal{D}_{\text{train}}$.

---

[4]To compute this curve, we split the target sequence probability $p_L(\boldsymbol{x})$ range into $N$ equally sized bins (by probability range). We report the mean estimation error for each bin with $> 10$ sequences. We additionally compute 95% confidence intervals with $10,000$ bootstraps for each mean, resampling $n$ equal to the number of sequences in the given bin.

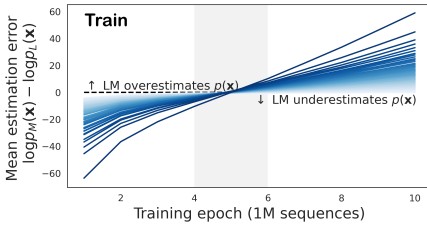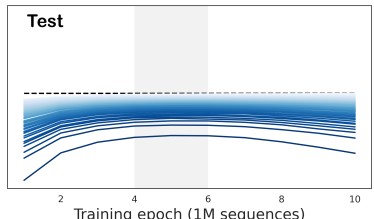

Figure 3: Mean model estimation error by training epoch (GPT2-medium). Each line denotes the mean estimation error for a 50th of the sequences; darker lines represent less probable sequences. The shaded area denotes the area in which validation cross-entropy reaches a minimum.

## 4.2 ESTIMATION ERROR ACROSS TRAINING TIME

To understand the training dynamics underlying the previously reported underestimation, we compute model probability estimates on subsets of $\mathcal{D}_{\text{train}}$ and $\mathcal{D}_{\text{test}}$ at the end of each training iteration $i$. Once computed, we sort each set of probability tuples $\mathcal{S}_{\text{train}}^{(i)}$ and $\mathcal{S}_{\text{test}}^{(i)}$ by their target sequence probabilities $p_L(\boldsymbol{x})$, and split the probability tuples into 50 equally-sized bins. We plot estimation curves in Figure 3: Each curve represents a 50th of the sequences, with darker curves denoting estimation error for sequences with lower target probabilities (rarer sequences). At any given point, then, the distance between estimation curves represents the degree to which estimation error is dependent on the target probability of the sequence.

Figure 3 left visualizes underestimation error for sequences seen in training. Around the fifth epoch, estimation error for train sentences converges to zero, that is $(p_L(\boldsymbol{x}) \approx p_M(\boldsymbol{x}))$, indicating that GPT2-medium is able to almost perfectly recover the target probabilities of training sequences no matter their target probability. At the same time, this convergence happens almost simultaneously for all sequences, indicating that a complete reduction in error during training occurs throughout the entire range of target sequence probabilities.

Figure 3 right visualizes GPT2-medium model's performance on a separate set of test sequences. First, unlike for $\mathcal{D}_{\text{train}}$, estimation error for $\mathcal{D}_{\text{test}}$ does not converge to zero, meaning that even when the model has perfectly recovered the target probability of train sequences, the target probabilities for test sequences remain underestimated. Second, in the case of $\mathcal{D}_{\text{train}}$, the difference between estimation curves of different shades converges to zero, indicating that estimation performance becomes uniform across the distribution of train sequences. We do not see such behaviour in $\mathcal{D}_{\text{test}}$. Instead, the error curves remain at a relatively consistent distance from one another, indicating that the discrepancy in estimation error at different parts of the distribution is unchanging for sequences not seen during training.

## 4.3 ESTIMATION ERROR BY AMOUNT OF TRAINING DATA

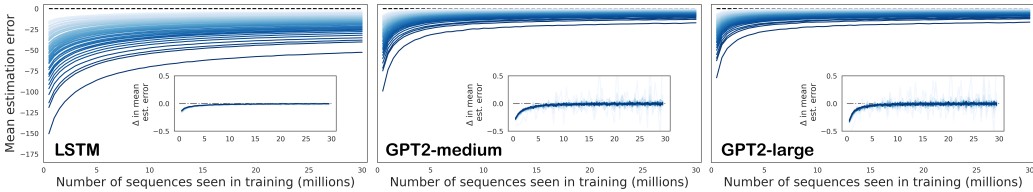

Figure 4: GPT2-medium, GPT2-large and LSTM trained on 30M sequences sampled from $p_L$. *Main plots*: Mean estimation error on test sequences as a function of the number of sequences seen in training. *Inset plots*: Relative change in mean estimation error of test sequences as a function of the number of sequences seen in training. In both cases, each line denotes estimation behaviour for a 50th of the test sequences; darker lines represent less probable sequences.

Our previous experiment trained languages models on a set of 1M sequences. A plausible explanation for the model's underestimation behaviour on unseen test sequences is therefore that the language model has not seen enough samples from the target distribution. Here we explore how

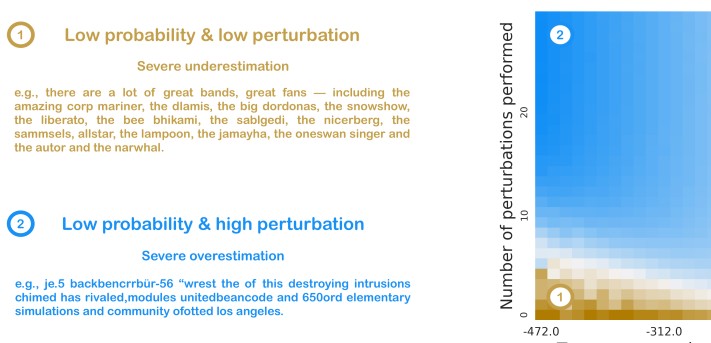

Figure 5: GPT2-medium estimation behaviour for 15M sequences in $\Sigma^*$ across two dimensions: **sequence rarity** (x-axis) and **degree of perturbation** (y-axis). The heat map is shaded based on estimation error severity; blue areas indicate overestimation, whereas brown areas indicate underestimation. We also include example sequences from two zones in this sequence space.

estimation error varies as a function of the amount of training data. We train GPT2-medium, GPT2-large and an LSTM model in the online "Ideal World" setting (Nakkiran et al., 2020) by sampling, at the beginning of each training iteration, a fresh set of 500,000 sequences from $p_L$, and training $M$ on this sample. Doing so for 60 iterations, we obtain LMs which have been trained on 30 million sequences. We compute model estimation error on $\mathcal{D}_{\text{test}}$ at the end of each iteration $i$. Figure 4 visualizes underestimation error throughout training for these LMs. We again split test sequences by their true probability, with darker lines denoting estimation trends for less probable target sequences.

The estimation curves in Figure 4 suggest that while increasing the amount of data in training initially leads to lower estimation error, this reduction eventually asymptotes. In the insets of Figure 4, we visualize the relative change in mean estimation between epochs $i-1$ and $i$. Relative change in estimation error eventually fluctuates around 0 (minimal change) for the majority of the distribution. Comparing architectures, we find that the Transformer is significantly more efficient at reducing mean estimation error throughout the distribution.

## 4.4 WHERE DID THE PROBABILITY MASS GO?

In the previous section, we saw that even when increasing the amount of training data, $p_M$ consistently underestimates the probabilities of sequences sampled from the tail of $p_L$. At the same time, we did not find that $p_M$ overestimated sequences in the head of $p_L$. Under the assumption that $p_M$ is a proper probability distribution,[5] that is, $\sum_{\boldsymbol{x} \in \Sigma^*} p_M(\boldsymbol{x}) = 1$, these findings suggest that there exists sequences in $\Sigma^*$ whose probability is overestimated by the model. In this section, we investigate where this probability mass went.

To do so, we compute model estimation error on perturbed sequences from $p_L$—sequences in $\Sigma^*$ which are increasingly far away from the high-probability zones in $p_L$. We build a corpus of perturbed sequences by recursively applying 30 random perturbations to each sequence $\boldsymbol{x} \in \mathcal{D}_{\text{test}}$. Formally, the set of sequences at perturbation step $i$ can be expressed as: $\mathcal{D}_{\text{perturbed}}^{(i)} = \{\text{PERTURB}(\boldsymbol{x}) \mid \boldsymbol{x} \in \mathcal{D}_{\text{perturbed}}^{(i-1)}\}$ where PERTURB$(\boldsymbol{x})$ is a function which returns a novel perturbed version of $\boldsymbol{x}$, and $\mathcal{D}_{\text{perturbed}}^{(0)} = \mathcal{D}_{\text{test}}$. Sequence perturbation operations are shown in Table 3. While it is possible that these operations produce other well-formed strings, we expect this to be a relatively rare outcome. We score each of these 15M sequences under both the target generative model $p_L$ and the LM $p_M$. Note that we use as LM the GPT2-medium model from the previous section (trained on 30M sequences).

**Swap** two tokens in $\boldsymbol{x}$.
**Delete** a token from $\boldsymbol{x}$.
**Insert** a token from $\Sigma$ at a position in $\boldsymbol{x}$.
**Substitute** a token in $\boldsymbol{x}$ with a token from $\Sigma$.

Table 3: PERTURB$(\boldsymbol{x})$ randomly applies one of these perturbations to $\boldsymbol{x}$.

---

[5]See Welleck et al. (2020a) for discussion on consistency in the context of neural language model decoding.

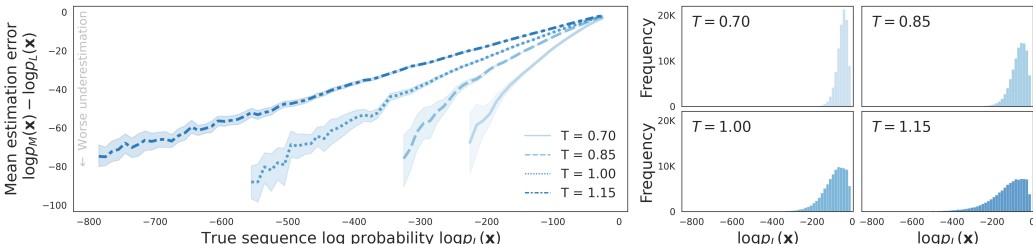

Figure 6: Model estimation error on test sequences as a function of target sequence probability for three different artificial languages. Each line visualizes estimation error for a GPT2-medium model trained to model a language with a specific softmax temperature parameter $T$.

Figure 5 visualizes GPT2-medium's mean estimation error for these 15M sequences across two dimensions. On the x-axis, we plot the target probability of the sequence under $p_L$, and on the y-axis, the number of perturbations performed on the sequence. For example, the bottom-left corner (1) of Figure 5 visualizes estimation behaviour for rare sequences sampled directly from $p_L$, whereas the top-left corner (2) visualizes estimation error sequences which are equally rare, but which have been perturbed up to 30 times.

Figure 5 offers a nuanced characterization of underestimation behaviour. The brown area on the bottom of the figure re-states the underestimation findings of the previous section. When increasing the number of perturbations performed, however, we begin entering into a space of sequences which are at first well-estimated by $p_M$ (the white areas) but then are quickly overestimated by $p_M$ (the dark blue areas), confirming that there are indeed sequences in $\Sigma^*$ which are overestimated by the language model. Furthermore, these findings suggest that the tail of rare events defined by the language model does not match the tail of the artificial language—the rare events typical in $p_L$ are under-represented in $p_M$ in favour of other sequences in $\Sigma^*$. See Section A.1.4 for experiments finding that random sequences from $\Sigma^*$ are also overestimated by $p_M$.

## 4.5 Modulating the Shape of the Target Distribution

Up to this point, the target artificial language $p_L$ was given as the distribution induced by an ancestral sampling scheme with softmax $T = 0.85$ from the generative model $L$. In the previous section, we saw that $p_M$ placed excess probability mass on areas in $\Sigma^*$ with low-probability under $p_L$. Here we modulate the shape of the sequence space defined by $p_L$ to investigate how estimation error varies with respect to systematic interventions in the target distribution. To adjust the way that $p_L$ allocates probability mass over $\Sigma^*$, we control the entropy of the conditional distributions at each generation step $t$ by dividing the pre-softmax logits by a temperature value $T$.[6]

We visualize the effects of $T$ on the shape of the distribution in the left of Figure 6. By increasing the value of $T$, we increase the entropy of the distributions over next tokens, which in turn, spreads probability mass across a larger number of sequences in $\Sigma^*$. We define four artificial languages with varying $T$ and train GPT2-medium on an ancestral sample of 1M sequences from each of these artificial languages. Figure 6 visualizes model estimation error by true sequence probability for each model. We find that models trained on languages with increased entropy perform comparatively better than models trained on low entropy languages. Estimation error for models trained on languages with greater $T$ is less severe, and this holds throughout nearly all target sequence probabilities. These results indicate that neural LMs provide a more accurate approximation of target distributions which spread mass more uniformly across $\Sigma^*$.

## 5 Related Work

This paper contributes to recent work investigating the properties of the distributions defined by LMs. Prior studies have focused on exploring (Takahashi & Tanaka-Ishii, 2019; 2017) and develop-

---

[6]Formally, for the $i$th component of the length $K$ pre-softmax logits $\boldsymbol{x}$, this operation is given as:
$$\text{SOFTMAX}(\boldsymbol{x})_i = \frac{\exp(\frac{x_i}{T})}{\sum_{j=1}^{K} \exp(\frac{x_j}{T})}$$

ing frameworks (Meister & Cotterell, 2021) to better understand whether the large-scale statistical tendencies of natural language, such as Zipf's law (Zipf, 1949), are captured by LMs. We take a more fine-grained approach, proposing a methodology which draws off of instance-level evaluation schemes (Zhong et al., 2021) and the experimental control afforded by artificial corpora (White & Cotterell, 2021; Papadimitriou & Jurafsky, 2020). Indeed, closely related to our work is Kulikov et al. (2021)'s, in which artificial corpora produced by generative models were used to explore mode recovery in neural language modeling. Our analysis exploring the overestimation of ill-formed sequences extends previous findings on locally normalized conditional models assigning arbitrary probability mass to unlikely sequences (Andor et al., 2016; Goyal et al., 2019; Lafferty et al., 2001), neural LMs assigning high likelihood to sequences with repetitions (Welleck et al., 2020b), the consistency of decoding algorithms (Welleck et al., 2020a), and on machine translation models placing significant probability mass on the empty sequence (Stahlberg & Byrne, 2019).

We additionally contribute to the body work seeking to characterize and adapt neural model performance on rare or novel examples and classes (Horn & Perona, 2017; Bengio, 2015). In the context of language modeling, Lent et al. (2021) explored performance on under-resourced languages, whereas Oren et al. (2019) did so on under-represented domains in training corpora. McCoy et al. (2021) introduced analyses to assess sequential and syntactic novelty in LMs. Focusing on the word frequency distribution, Dudy & Bedrick (2020) found that LMs under-perform when less frequent examples are encountered at test time. In the classification setting, various approaches have been proposed to help alleviate class imbalance in the data distribution, such as data augmentation (Sagawa et al., 2020) or the transfer of knowledge from high-frequency classes to infrequent ones (Ouyang et al., 2016; Zhu et al., 2014; Chen et al., 2021). Prior to the current neural paradigm (Bengio et al., 2003), multiple approaches have been proposed to deal with the heavy-tail, such as smoothing and back-off approaches in statistical $n$-grams (Chen & Goodman, 1999) and two-stage Bayesian approaches (Goldwater et al., 2006).

## 6 CONCLUSION

Emerging as a result of a language user's ability to produce and comprehend novel expressions, the heavy-tail of rare events is one of the fundamental features of distributions in natural language. In this work, we introduce a controlled methodology to evaluate instance-level LM performance on this set of individually rare but collectively frequent events. We use generative models trained on natural language corpora to define a set of artificial languages for which we can exactly compute the probability of sequences. Training LSTM and Transformer LMs on sequences sampled from these artificial languages, our analysis compares the probability estimates given to sequences by the LMs to the target probabilities of sequences under the artificial language.

Our results indicate that neural LMs systematically under-represent sequences in the tail of the target distribution, even when increasing the amount of the training data. Investigating where this probability mass went, our perturbation experiments reveal that neural LMs do not tend to overestimate the head of the distribution, but rather overestimate the probability of sequences outside those typical in the target distribution. Comparing model performance on target distributions with varying properties, we find that neural LMs tend to provide more accurate approximations of distributions with greater entropy. Interpreted together, these results indicate that autoregressive neural language models have a tendency to spread probability mass too uniformly across the space of possible sequences.

Finally, we would like to acknowledge that we do not know the degree of structural difference between our Transformer-generated ground-truth distributions and the distributions of actual natural languages. It is likely that the distribution defined by our ground truth models is less structured than the distribution of a natural language. Therefore, it is possible that some systematic difference between natural language distributions and our ground-truth distributions may affect our results to a certain degree. That being said, our experiments in Section 4.5 suggest that it may actually be easier for neural LMs to learn less structured distributions, and we expect the task of recovering a ground-truth distribution to be made easier when the target distribution and LM are in the same model class. Nevertheless, future work should seek to conduct similar experiments using ground-truth distributions with more explicit structure.

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

## A    APPENDIX

### A.1    ADDITIONAL EXPERIMENTS

#### A.1.1    PRE-TRAINED GROUND-TRUTH MODEL

In our previous experiments, our ground-truth model was given as a Transformer language model trainined on 1.5M sequences from the OpenWebText corpus. Here we explore underestimation behaviour when the ground-truth distribution is given by a pretrained GPT2-medium model fine-tuned on 1.5M sequences from the OpenWebText corpus. We sample from $p_L$ with softmax $T = 1.00$.

Analogously to the experiment in Section 4.1, we train a randomly-initialized GPT2-medium model on 1M sequences sampled from the fine-tuned model. In the center of Figure 7, we visualize mean model estimation error for $50,000$ test sequences as a function of true sequence probability. Similarly to our experiments in Section 4.4, we find that the LM underestimates the probability of the majority of sequences, and does so more severely for less probable sequences. Note that this LM obtains a test perplexity of 67.97.

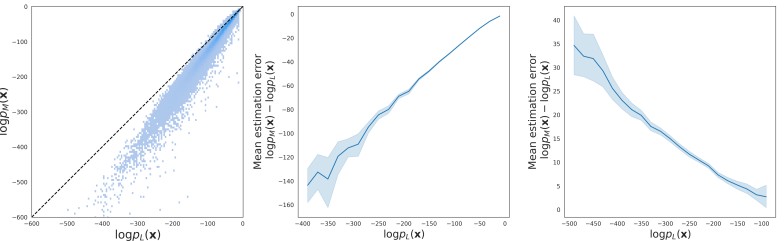

Figure 7: *Left*: Joint histogram of sequence probability estimates for test sequences. *Center*: Mean model estimation error by true sequence probability for test sequences. *Right*: Mean model estimation error by true sequence probability for sequences randomly sampled from $\Sigma^*$.

Finally, to ensure that our findings regarding the overestimation of ill-formed sequences hold, we compute model estimation error on random sequences sampled from $\Sigma^*$ (see A.1.4 for details on how these sequences are constructed). Figure 7 center visualizes mean model estimation error as a function of target sequence probability. We find that $p_M$ overestimates the majority of ill-formed sequences, indicating that these findings hold when the ground-truth distribution is defined using a pretrained model.

#### A.1.2    UNTEMPERED (T = 1.00) GROUND-TRUTH DISTRIBUTION

In Sections 4.1 to 4.4, we conducted all experiments on an artificial language defined by ancestral sampling scheme with $T = 0.85$. In Section 4.5, we saw that the underestimation findings held regardless of $T$. To provide further evidence that these results hold for other values of $T$, we conduct similar experiments as in Section 4.3 with an artificial language $p_L$ defined by an ancestral sampling scheme with $T = 1.00$. Specifically, we train GPT2-medium and an GPT2-large on a total of 30M sequences sampled from $p_L$, and we compute model estimation error on a set of withheld test sequences at each training iteration.

#### A.1.3    PRE-TRAINED MODEL ESTIMATION ERROR

In Section 4.3, we explored how estimation error varies as a function of the amount of training data, finding that while increased data weakens estimation error, the underestimation behaviour persists. As an alternative way to explore how underestimation varies with increasing data, we fine-tune Huggingface's pre-trained GPT2-small, -medium and -large models on the set of 1M sequences used in Section 4.1. Computing estimation error on a set of unseen test sequences, we find, analogously to our experiments on models trained from scratch, that pre-trained models underestimate the probability of the majority of sequences sampled from the target distribution, and do so more severely for rarer sequences (we visualize this in Figure 9). Furthermore, in Figure 10, we increase the amount

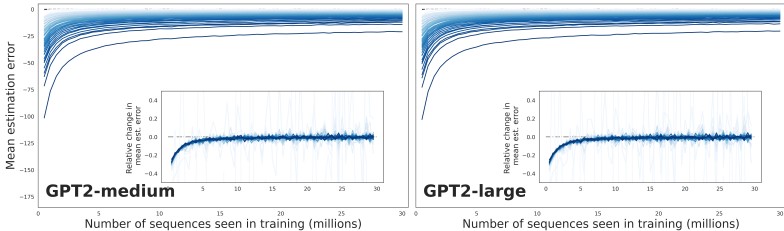

Figure 8: GPT2-medium and GPT2-large trained on 30M sequences sampled from $p_L$ with $T = 1.00$. *Main plots*: Mean estimation error on test sequences as a function of the number of sequences seen in training. *Inset plots*: Relative change in mean estimation error of test sequences as a function of the number of sequences seen in training. In both cases, each line denotes estimation behaviour for a 50th of the test sequences; darker lines represent less probable sequences.

of fine-tuning data substantially, and we plot test sequence estimation behaviour at the end of each epoch for a pre-trained GPT2-medium model. Once again, increased training data lessens but does eliminate the underestimation behaviour.

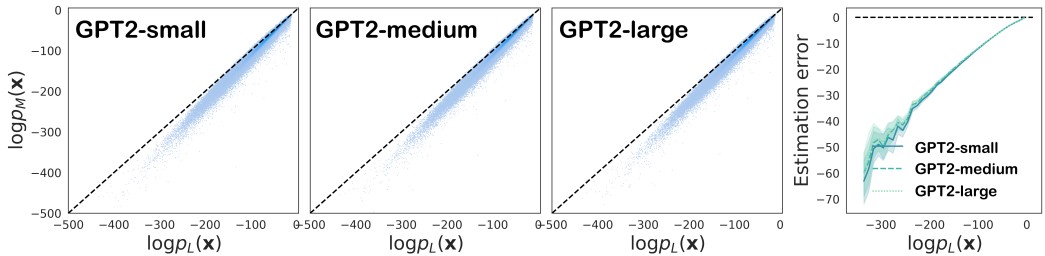

Figure 9: Test sequence probability estimates given by pretrained neural LMs fine-tuned on 1M sequences sampled from $p_L$. *Three left-most figures*: The joint histograms of sequence probability estimates. The dotted line denotes the cases in which the model's estimates perfectly align with the target probability; shading to the right of this line denotes underestimation. *Right-most figure*: Mean sequence estimation error by target sequence probability.

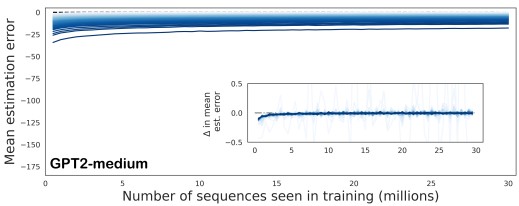

Figure 10: Pre-trained GPT2-medium fine-tuned on 30M sequences sampled from $p_L$. *Main plots*: Mean estimation error on test sequences as a function of the number of sequences seen in fine-tuning. *Inset plots*: Relative change in mean estimation error of test sequences as a function of the number of sequences seen in fine-tuning. In both cases, each line denotes estimation behaviour for a 50th of the test sequences; darker lines represent less probable sequences.

### A.1.4 ALTERNATIVE PERTURBATIONS

In Section 4.4, we study where the language model's underestimated probability mass went by computing model estimation error on perturbed sequences. We obtain a set of perturbed sequences by (i) sampling a sequence from $p_L$ and then (ii) recursively perturbing this sequence according to the perturbations provided in Table 3.

This method provides us with sequences which are increasingly far away from high-probability zones under $p_L$. However, it does so with initial sequences sampled directly from $p_L$, and as a result, produces strings which are edit-adjacent to the high-probability zones (under $p_L$) in $\Sigma^*$.

To ensure that our results hold in other low-probability subspaces, we conduct an analogous experiment on sequences randomly sampled from $\Sigma^*$. We sample a sequence from $\Sigma^*$ by first sampling a sequence length $l$ from a Poisson distribution with $\lambda = 10$. Given this length, we sample $l$ tokens from $\Sigma^*$ and concatenate all tokens to form a sequence. We then score the sequence under both $p_L$ and $p_M$, and compute model estimation error. We use as artificial language $p_L$ GPT2-medium with $T = 0.85$ and we use as language model $p_M$ a GPT2-medium model trained on 30M sequences ancestrally sampled from $p_L$.

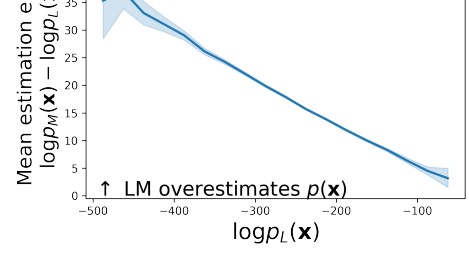

Figure 11: Mean model estimation error by true sequence probability for sequences randomly sampled from $\Sigma^*$.

Figure 11 visualizes mean estimation error as a function of the ground-truth probability of the sequence. Similarly to all other perturbation experiments, we do indeed find that $p_M$ overestimates these sequences, regardless of their true sequence probability.

### A.1.5 ESTIMATION ERROR BY SEQUENCE LENGTH

Autoregressive neural language models decompose the joint distribution $p(\boldsymbol{x})$ over sequences into a series of conditional distributions $p(x_i \mid x_{<i})$. Generating a sequence of length $n$, then, involves estimating $n$ conditional distributions. Since the probability of a sequence is inversely correlated with its length, our findings that estimation error is worse for rarer sentences may be explained by compounding errors.

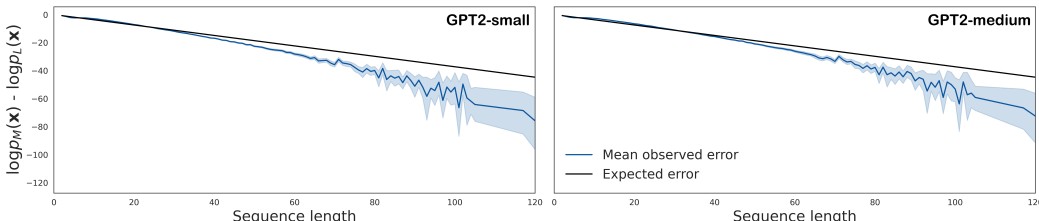

Figure 12: Expected and observed estimation error by sentence length for GPT2-small and GPT2-medium.

To test this claim, we ask whether the observed estimation error is worse than would be expected if it was due to error compounding. Specifically, in Figure 12, we plot the expected model estimation (black) and the mean observed error (blue) by sequence length. Expected estimation error for sequence length $n$ is computed by multiplying the average token level estimation error by the sequence length, i.e., $n\bar{\epsilon}$, where

$$\bar{\epsilon} = \frac{1}{|\mathcal{D}|} \sum_{\boldsymbol{x} \in \mathcal{D}} \frac{1}{|\boldsymbol{x}|} \sum_{i=1}^{|\boldsymbol{x}|} \log p_M(x_i \mid x_{<i}) - \log p_L(x_i \mid x_{<i})$$

Observed estimation error is computed as the mean estimation error for test sentences of length $n$. Note that the shaded areas around this curve denote the 95% bootstrapped confidence intervals for this mean. As shown in Figure 12, observed estimation error for both GPT2-small and GPT2-medium is more severe than expected estimation error as we increase sentence length. This suggests that estimation error for longer (and typically rarer) sequences is not solely due to error compounding.

## A.2 EMPIRICALLY MEASURING THE LNRE ZONE

In section 2, we formally defined the LNRE zone as the range of values of $N$ for which there is non-zero probability of sampling a novel event at the $N + 1$th draw. Here we introduce a frequentist estimator for this probability. This in turn allows us to empirically verify if a sample exists in the LNRE zone.

### A.2.1 ESTIMATING THE POTENTIAL PRODUCTIVITY

Suppose we have a set of $N$ events $\mathcal{D} = \{\boldsymbol{x}_1, \ldots, \boldsymbol{x}_N\}$ drawn from some generative process $\phi$. Given $\mathcal{D}$, we aim to obtain an empirical estimate for the potential productivity $\hat{\mathcal{P}}_N$: the amount of probability allocated to unseen events as a function of $N$. We can do so using the Good-Turing estimate for the probability of an event given its frequency (Good, 1953).

Specifically, let $f(\boldsymbol{x}, \mathcal{D})$ be a function which returns the frequency of the event $\boldsymbol{x}$ in $\mathcal{D}$. Let $N_m$ denote the number of types (unique events) in $\mathcal{D}$ for which $f(\boldsymbol{x}, \mathcal{D}) = m$. Good-Turing says that for large $N$, the probability of the event $\boldsymbol{x}$ given that it has occurred with frequency $m$ in our sample $\mathcal{D}$ of size $N$ is equal to

$$\hat{P}(\boldsymbol{x} \mid f(\boldsymbol{x}, \mathcal{D}) = m) = \frac{(m+1)}{N} \frac{N_{m+1}}{N_m} \tag{3}$$

To obtain an estimate for $\mathcal{P}_N$, we set $m = 0$:

$$\hat{\mathcal{P}}_N = N_0 \hat{P}(\boldsymbol{x} \mid f(\boldsymbol{x}, \mathcal{D}) = 0) = \frac{N_1}{N} \tag{4}$$

which states that the total amount of probability mass allocated to unseen events is equal to the proportion of events which occurred only once (*hapax legomena*) in $\mathcal{D}$. This quantity is known as the *potential productivity* of a linguistic process (Baayen, 2009; 2001; 1994).

### A.2.2 THE LNRE ZONE IN OPENWEBTEXT

We apply this method to a subset of OpenWebText, a popular language modeling corpus. In Figure 13, we plot the the empirical estimate of $\mathcal{P}_N$ as function of the sample size $N$ for $n$-grams sampled from a subset of OpenWebText. Particularly for $n$-grams with $n \geq 3$, we find that there is significant probability of sampling a previously unseen event, even for $N > 10,000,000$.

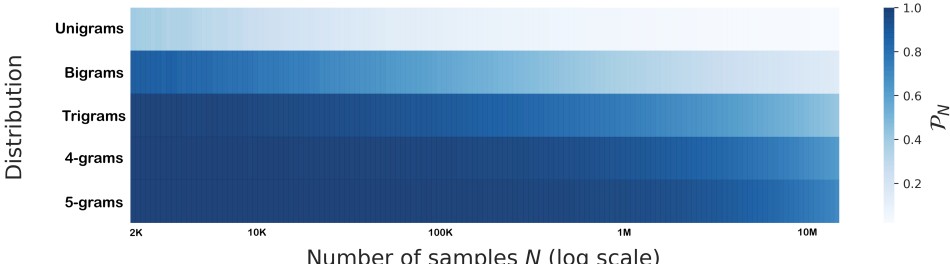

Figure 13: The probability of sampling a novel item (potential productivity $\mathcal{P}$) as a function of sample size $N$. Many distributions in natural language are characterized by a potentially unbounded amount of novelty.

## A.3 MODEL PERPLEXITY VALUES

In this section we report relevant perplexity values for all models used. For each model, we report mean perplexity across sentences drawn from (i) the validation set generated by the artificial language they attempt to model and (ii) the OpenWebText corpus. While we include the perplexity values for our models on sentences of English, this is not to claim that our ground-truth models are meant to define a distribution which closely resembles the distribution of English.

| Model | mean $PP$ (val) | mean $PP$ (eng) |
|---|---|---|
| LSTM | 36.53 | 158.31 |
| GPT2-small | 26.66 | 129.84 |
| GPT2-medium | 25.42 | 132.56 |
| GPT2-small (pretrained) | 21.02 | 53.79 |
| GPT2-medium (pretrained) | 20.87 | 48.42 |
| GPT2-large (pretrained) | 20.90 | 47.30 |
| LSTM (increased data) | 35.01 | 149.02 |
| GPT2-medium (increased data) | 21.60 | 94.03 |
| GPT2-large (increased data) | 21.32 | 91.28 |
| GPT2-medium (ground-truth model) | - | 73.79 |

Table 4: $PP$ on validation set generated by the artificial language the LM attempts to model (val), and on real sentences sampled from OpenWebText (eng).

| **Softmax** $T$ | mean $PP$ (val) | mean $PP$ (eng) |
|---|---|---|
| 0.70 | 11.61 | 223.60 |
| 0.85 | 25.42 | 132.56 |
| 1.00 | 75.30 | 106.21 |
| 1.15 | 290.69 | 106.29 |

Table 5: GPT2-medium $PP$ on validation set generated by the artificial language the LM attempts to model (val), and on real sentences sampled from OpenWebText (eng) for models used in Section 4.5

## A.4 LANGUAGE SAMPLES

| $\boldsymbol{x}$ | $\log p_L(\boldsymbol{x})$ |
|---|---|
| he was a complete east end player . | $-26.399$ |
| a former harvard university graduate told cnn that in recent weeks , u.s. intelligence officials have begun to gather evidence that trump 's campaign colluded with russia to influence the election . | $-61.846$ |
| in the current study , we examined whether participants in the study performed more or less " active " in weight loss -lrb- p = 0.05 -rrb- . | $-51.6566$ |
| you , the one that is the republican candidate , who is taking over the senate and government as a democrat and who is a bipartisan democrat , and you 've got to be able to get that done . | $-92.703$ |
| since the 1970s , the city has been in the midst of a landmark urban pride . | $-44.9523$ |

Table 6: Sequences ancestrally sampled from the artificial language generated by GPT2-medium with softmax $T = 0.70$.

| $\boldsymbol{x}$ | $\log p_L(\boldsymbol{x})$ |
|---|---|
| " i have no doubt that 's a big part of any kind of campaign machine , " he told al jazeera . | $-54.5951$ |
| and it 's not a very good idea to work with claims in comics and film novels . | $-59.7626$ |
| according to the new york times , susan macmahon , 54 , and her husband , elizabeth bailey , were in the vehicle with their son , aged between 12 and 19 . | $-86.5217$ |
| anticipating experience | $-21.7095$ |
| this is the reason i 'd like to take advantage of these ideas in an attempt to transform my life . | $-53.1399$ |

Table 7: Sequences ancestrally sampled from the artificial language generated by GPT2-medium with softmax $T = 0.85$.

| $\boldsymbol{x}$ | $\log p_L(\boldsymbol{x})$ |
|---|---|
| beautiful bacon cookies | $-19.8038$ |
| " this accident , despite most of its life , is dependent on one of the most precious bodies on this planet , which is comparable to that on mostrughenai mountains . | $-142.4362$ |
| it 's not fair to say that we 're not in a position to permit same - sex marriages , and all the same issues that are presented on regular basis . " | $-88.9437$ |
| from video -lrb- without editing -rrb- you can see original images produced at the official website , competitors , and event prices . | $-100.3791$ |
| consumer electronics have become a perfect fit for bm 's ultra - low - cost turn . | $-73.0217$ |

Table 8: Sequences ancestrally sampled from the artificial language generated by GPT2-medium with softmax $T = 1.00$.

| $\boldsymbol{x}$ | $\log p_L(\boldsymbol{x})$ |
|---|---|
| zy brace gym fingerprint – bom jerozo dell ' | $-94.0307$ |
| shortly thereafter , will be a buoyant as the dynamo bust series starts one . | $-90.5994$ |
| disjitor doom efeco became a gothicrevolution manager ofby city library for marvel studios . | $-143.6075$ |
| earlier this week chicago 's writer andrew o't text ' american catholics to interested readers the holiday kingsburyobee -lrb- expiration -rrb- at red for now atmotionweism race and bachelor whitney university , register as 1852 jim banner of airbus beer and of course smiled at us in formula 1 where he tells friends by name , " finnish catholics : the encryptings on robot - type mothers . | $-493.9313$ |
| middleware in germany | $-27.3411$ |

Table 9: Sequences ancestrally sampled from the artificial language generated by GPT2-medium with softmax $T = 1.15$.

