# OpenReview forum: "Evaluating Distributional Distortion in Neural Language Modeling"
_ICLR.cc/2022/Conference — ICLR 2022 Poster_

### Official Review · Reviewer_Msuw · 2021-11-01

**Correctness:** 3
**Technical Novelty And Significance:** 2
**Empirical Novelty And Significance:** 3
**Recommendation:** 8
**Confidence:** 3

**Main Review:**

* Strengths: A very well written and clear paper, addressing a focussed question through a well-designed set of experiments on synthetic data allowing to precisely control experimental conditions.

* Weaknesses: No major weakness, but some mild issues and areas for improvement, see the comments and questions below.

* Comments and Questions:

    * (Minor Comment) I did not find the description of the LNRE Zone in section 2 to bring much to the discussion. In particular, characterizing this zone through the condition that $\mathcal{P}_N$ is not __null__ is almost empty of content for natural language, because it would in particular mean that we could precisely delimit what constitutes a possible "natural" sentence, and that the LNRE Zone should cover all such sentences. Here the formal description as given does not clarify anything.
    * (Question) In some of your experiments, you train a $P_M$ GPT2-medium model on 30M sentences sampled from the $P_L$ GPT2-medium model, that is, $P_M$ here uses the same architecture as $P_L$ (if I understand correctly). However, the $P_M$ model still underestimates the probabilities relative to $P_L$. Do you think this is a symptom of 30M sentences being too few, or more a symptom that $P_M$ is stuck in some local minimum during the optimization ? After all, (again, if I understand the setup correctly), using the same architecture, $P_M$ would make it possible, in principle, to just use the same parameters as $P_L$, and then the mismatch could not occur. Perhaps this would be worth a note in the text?
    * (More important Question/Comment) In section 4.4, you perform some perturbations, and observe that they lead to higher $P_M$ than $P_L$. First, I think it would be worth defining these perturbations in more detail than what you do. Second, you do not discuss your choice for these perturbations nor analyze why they have (relatively) high probability relative to $P_M$. However, the fact that, for instance, swapping two words or deleting a word in a natural sentence is not severely penalized by $P_M$ is an interesting but specific fact, which may have to do with a special tolerance of $P_M$ for such operations, rather than for others. To give one example, if the perturbations consisted solely in replacing a word in the natural sentence by a _rare_ word in the training set, would you observe such a difference between $P_M$ and $P_L$? Overall, could you discuss this aspect of your experiments?



**Summary Of The Paper:**

The paper conducts experiments to evaluate whether neural sequence models such as LSTMs and Transformers are able to correctly assess the probability of rare sentences, which collectively constitute a large probability mass in natural language productions (heavy-tail phenomenon). In order to do so, it performs experiments in a controlled synthetic environment where a first language model $P_L$ is trained on a corpus of natural sentences, and a second model $P_M$ is trained to emulate the first model. The authors observe that the second model systematically tends to underestimate the probability of rare $P_L$ sentences, the more so the rarer such sentences are, and show that some artificially corrupted sentences tend to receive higher probability from $P_M$ than from $P_L$, partly explaining where the missing probability mass over rare well-formed sentences went.



**Summary Of The Review:**

A nice experimental paper addressing the difficulty of neural LMs to approximate the probability of rare events from an underlying "teacher" LM. The paper could be improved by more analysis and discussion of some of the results.

*After authors' response*: thank you for your detailed answers to my questions and for your additions to the paper and to experiments. I aim raising my score and hope the paper will get accepted.

---

> ### Author Response · Authors · 2021-11-15
> **Response to Reviewer Msuw**
>
> Thank you for the thoughtful and useful comments. Please see our comments below and the revised version of the manuscript. All significant changes are in blue.
>
> **Comment on LNRE zone discussion**: We appreciate this point. The formal description was simply meant to make explicit the connection between the massive amount novelty that language models have to deal with and the productivity of language. We have added a sentence clarifying this point to the paper.
>
> **Question about local minimum and increasing data**: This is a very good point. Our results highlight that underestimation asymptotes within 30M sequences. This result suggests that larger amounts of training data have relatively minimal effects on underestimation error. Nonetheless, it is still likely that $P_M$ is stuck in a local minimum during optimization. One possible way to investigate whether this is true is to train a larger model on sentences sampled from $P_L$. If the larger model provides a better fit, then perhaps there are optimization issues (related work indicates that overparameterization may be more beneficial than increasing the amount of data [1]). We are currently training a $P_M$ GPT2-large model on sentences sampled from the $P_L$ GPT2-medium model to see whether the large model provides a better fit. Our initial results (after 15M sequences) suggest that GPT2-large provides a marginally better fit ($P_M$ GPT2-med obtains perplexity of 22.4411 after 15M sequences, whereas $P_M$ GPT2-large obtains 22.3180) but still suffers from underestimation. We will add these considerations in the main text once our experiments are completed. Another experiment which may help elucidate whether the model is in a local minimum would be to deploy multiple training runs with different random seeds. If all training runs converge to the same test PPL score, then we can take this evidence that we are not in a single local minimum.
>
> **Comment/Question on perturbation experiments**: This is a good point. It could be the case that the specific perturbations we chose may be particularly tolerated by our student models. However, since our teacher and student models are in the same model class, then we might expect sequences perturbed using these edits to also be assigned relatively high probability by the teacher, which they are not.
>
> The goal of the perturbations experiments was to examine the probability of sequences in Σ* outside of high-probability subspaces under $P_L$ . Based on your comment, we have added an additional experiment to the Appendix which examines the probability of randomly sampled sequences from Σ* under $P_L$  and $P_M$ . Specifically, we sample a sequence length l from a Poisson distribution, then uniformly sample tokens from the model’s vocabulary. These experiments show that $P_M$ tends to overestimate the probability of these sequences as well. This indicates that the overestimation behavior of $P_M$  is likely attributable to a general tendency to spread probability mass more uniformly on Σ* rather than a consequence of our specific edit operations.
>
> Nevertheless, we agree that examining other edit operations would also be interesting and we would like to hear if you have any suggestions for further approaches to this problem. In particular, we would like to add the edit you suggested, but need more clarification regarding how this perturbation differs from the substitution perturbation defined in Table 3. Finally, we have added more information in the paper about how our perturbed sentences are constructed.
>
>
> Reference: [1] https://arxiv.org/pdf/2002.11794.pdf

---

> > ### Comment · Reviewer_Msuw · 2021-11-28
> > **Thank you for your responses --- I raised my score and hope the paper will get accepted**
> >
> > Thank you for your detailed answers to my questions and for your additions to the paper and to experiments. I aim raising my score and hope the paper will get accepted.

---

### Official Review · Reviewer_AtM6 · 2021-11-03

**Correctness:** 3
**Technical Novelty And Significance:** 3
**Empirical Novelty And Significance:** 2
**Recommendation:** 8
**Confidence:** 4

**Main Review:**

The paper overall is exceptionally well-written and clear and tables and plots are beautiful and clear---multiple times did I go "ooh that's a nice way of laying it out." Kudos! The abstract tells the entire story and the paper substantiates the claims nicely making what is shown feel easy and obvious, which I consider a great feat.

Concerning content, I am equally enthusiastic about the idea and evaluation and agree that this can be a very interesting piece of evidence that nicely connects to a number of past threads of NLP research (as laid out throughout and in the nice related work section, though I don't think I know enough related literature to truly safely say that the paper is as novel as it claims to be). That said, there are three major methodological concerns that I would like to discuss through with the authors before recommending acceptance:

1. We do not know whether neural-LM-defined origin distributions are "strange"---or really, structurally different from actual human natural language distributions that we believe exist---in some systematic way that leads to the results in this paper. One potential intuition here is that a neural LM being a very imperfect model makes very erratic choices for what low-frequency/low-likelihood it assigns probability mass to---erratic choices that may be very hard to learn/emulate with the student LM that the authors train. Maybe real natural language distributions have more "sensible tails" that are actually not all that hard to learn and maybe not even underestimated. To be clear, this is a fundamental issue that the authors have no way of *resolving* easily under this paradigm and I think the paradigm is still a worthwhile idea! What I am asking is that this limitation is acknowledged and discussed with the same clarity that is afforded to the other content.

2. Relatedly, currently there is no way to know whether the trained models are actually reasonably trained, overall well-working models or not: the paper is missing test ppl/bpc for both the origin model (on English test data) and for the student model (both on the original English data and on test data sampled from the origin model). The authors make a good point that these scores are not as informative as what this paper presents, but the reason these are crucial is so we know whether the effects the authors find are a property of well-trained models or whether these models (through subpar training or another issue) are just poor models that have little connection to the models we actually use in reality. A secondary idea to easy this concern might be to actually use a pretrained GPT model as the origin model to inspire trust that that at least is a good model---I'm not sure why this road wasn't chosen unless it truly did not matter and this "home-made" model performs just as well, in which case, again, I would like to see some quantitative substantiation of that claim.

3. The paper is not really clear on how the sampling/teaching procedure and the tempering of the softmax interact: Footnote 2 claims pure (right?) ancestral sampling is used (as opposed to top-k/nucleus?), but then Footnote 4 says the softmaxes are all cooled to T=0.85, and yet Section 4.5 again implies that no tempering took place up to that point in the study. This is not only slightly confusing messaging, but I am worried that it may belie the promise of fair evaluation, especially because it isn't clear to me whether the tempering was accounted for in the calculation of probabilities as defined at the end of section 3.2. Specifically, if the probability of a sentence under the origin model does not take into account this local tempering, then it is no surprise that the student model learns to underestimate rare events because they just weren't sampled thanks to the locally cooled distribution! I would be somewhat reassured if the probability calculation in section 3.2 takes the tempering into account, but my understanding of the paper is that it currently fails to do so. As slight reassurance, Figure 5 does seem to tell us that even without any tempering (yes?) the described effect is visible, so I think this too should not be an issue that sinks the paper, but I do see it as a critical flaw the way it is right now.

Beyond those three, I see some minor concerns that may be worth addressing:
- "autoregressive neural language models" should also cite the Mikolov RNN paper, perhaps in place of one of the two GPT papers
- not to Schmidhuber, but citing only Melis et al. (2020) for LSTMs looks a little odd (citing Hochreiter & Schmidhuber or Sundermeyer et al. may be less surprising though I understand the desire to cite the *actual* model and training specification used and agree it should be present)
- concerning citations, Shiran Dudy & Steven Bedrick's work may be nice to also connect to (despite its relatively low visibility)
- it may be worth pointing out that the perturbations in section 4.4 may individually also make sentences *more* grammatical and likely and that is just a relatively rare outcome (I don't think much more than that needs to be said on that)
- I did not quite see what was "[p]erhaps unsurprisin[g]" in Section 4.5, but that's a minor nitpick
- Appendix A.1 measures events, okay... but are those words (as the hapax legomena theory usually assumes) or are they sentences (as I think this experiment does)? If they are sentences, we should not be surprised to see very high probabilities of seeing novel events, after all sentences are rarely the same (excluding formulaic language and "Resumption of the session" style sentences). I also would have liked to see some Poisson distributions or other distributions that were more interesting than uniform n-sided dice in Figure 6, but that is more wish than necessity.

**Summary Of The Paper:**

The paper proposes to examine the probability mass that a language model trained as usual places on frequent/likely and rare/unlikely sentences by learning from a distribution that is another neural language model itself so instance-level probabilities/NLLs of sentences can be compared. It turns out that all models in all tested circumstances *underestimate* the probabilities of rare events---and an ablation study shows that much of that missing mass can be found in sequences that are also rare and unlikely, but that weren't likely under the origin model.

**Summary Of The Review:**

Before author response:

My comparatively low score is a consequence of my concerns and very much given as a temporary score until we can discuss the three major concerns I have---if they are addressed satisfactorily, I anticipate raising my score significantly to champion the paper if needed as I found this paper a joy to read and thought-provoking in a good way (even if that lead to criticism).


After author response:

Thank you for all the clarifications and edits, they are greatly appreciated. I would strongly like to see this paper accepted.

---

> ### Author Response · Authors · 2021-11-15
> **Response to Reviewer AtM6**
>
> Thank you for the thoughtful and useful comments. Please see our comments below and the revised version of the manuscript. All significant changes are in blue.
>
> **Major concern 1) The properties of the tails in the ground-truth distribution**: This is a very good point. We agree that our lack of knowledge of the properties of the ground-truth distribution should be acknowledged and discussed. To this end, we added a paragraph at the end of the conclusion in our revision. We state it is likely that the distribution defined by our ground-truth models is less structured in the tail than the distribution of a natural language. We expect this to be the case since the main takeaway of our paper---that neural-LMs spread mass too uniformly over Σ*--- holds for our ground-truth models as well. That being said, we expect this lack of structure to make it easier for our student LMs to recover the distribution; our experiments in Section 4.5 indicate that underestimation increases when decreasing the entropy of the target distribution. Therefore, it is not unreasonable to expect student neural-LMs to underestimate more severely when trained on distributions with more explicit structure. Possible ways to test this claim include the use of grammars (e.g., a PCFG) as ground-truth distributions. However, something like a PCFG typically provides poor coverage of natural languages, and scoring sequences under these models can be costly. As a middle ground, one could guide the sampling of sentences from a neural LM such that they have high probability under an external grammar. We leave these experiments to future work.
>
> **Major Concern 2) Performance of ground-truth models**: Using a pre-trained GPT is an excellent suggestion. We have added an additional experiment to the Appendix which uses a pretrained GPT2-medium model as the ground-truth distribution. Specifically, we fine-tune a pretrained GPT2-medium model on 1.5M sequences sampled from the OpenWebText corpus. Training a randomly-initialized GPT2-medium model on generations from this pretrained model, we find that the student language model underestimates the probability of the majority of test sequences, and does so more severely for less probable sequences. Thus, these results are similar to those we saw for non-pretrained ground-truth distributions. Finally, we have also added the perplexity values for the models used in the paper to the Appendix.
>
> **Major concern 3) Sampling and scoring information**: Thank you for letting us know that our sampling procedure was not clear. We do indeed account for tempering in the calculation of the ground-truth probabilities. Additionally, all experiments prior to Section 4.5 are conducted for distributions induced by an ancestral sampling scheme with softmax $T=0.85$, as suggested by footnote 4. Note that we tempered the ground-truth distribution in an attempt to define an artificial language with more realistic entropy. Given the findings of our paper, it is likely that the distribution defined by our ground-truth models spreads mass more uniformly over Σ* than a natural language does. By tempering the distribution, we attempt to lessen this discrepancy. We have added clarifications in Sections 3.1, 3.2 and 4.5 to make these points clearer.
>
> As you noticed, the underestimation effect occurs whether the distribution is tempered or not. As further evidence of this, we have added a plot to the Appendix which shows GPT2-small and LSTM underestimation behaviour when fitting an untempered ($T=1.00$) ground-truth distribution. The trends are analogous to those found for $T=0.85$. For conceptual simplicity and clarity, we are also re-running all experiments prior to section 4.5 without any tempering. For completeness, in the final version of the paper, we will add results for all experiments when setting $T=1$.
>
> **Minor concerns**: Thanks for the great suggestions. We have added the relevant citations, and have added a mention of Shiran Dudy & Steven Bedrick's work in our related works section. We have also added a sentence mentioning that our perturbations may in rare cases produce more grammatical sentences, and have added an additional experiment to the Appendix which computes model estimation on random sequences in Σ*. We have removed the “perhaps unsurprisingly” from Section 4.5. Finally, the Appendix productivity measures are for sentences; and we agree that this is not a surprising fact, but we think it is nevertheless important in the context of language modeling. We will add distributions other than uniform dice, along with measures for words in the final version of the paper.

---

> > ### Comment · Reviewer_AtM6 · 2021-11-22
> > **Brief Reply**
> >
> > Thank you for your detailed reply and the updates to the manuscript---I raised my score and hope this paper gets in! :)

---

> ### Author Response · Authors · 2021-11-16
> **Response (2) to Reviewer AtM6**
>
> Dear Reviewer,
>
> We have just updated the manuscript with the finalized version of the experiments using a pretrained ground-truth model. Please see Section A.2. We fine-tuned a pretrained GPT2-medium model on 1.5M sequences sampled from the OpenWebText corpus, and trained a randomly-initialized GPT2-medium model on generations from this pretrained model. We find that the student language model underestimates the probability of the majority of test sequences. Furthermore, this underestimation is more severe for less probable sequences. Therefore, these results are similar to those we saw for non-pretrained ground-truth distributions. In fact, the underestimation seems to be more severe in this case.

---

### Official Review · Reviewer_V22X · 2021-11-03

**Correctness:** 4
**Technical Novelty And Significance:** 4
**Empirical Novelty And Significance:** 4
**Recommendation:** 8
**Confidence:** 5

**Main Review:**

**Interesting background**. The connections between productivity, low-probability sequences, and the limitations of perplexity were clearly written and interesting.

**Well-executed experiments and methodology**. The idea of using a language model as a ground-truth distribution was interesting and well-suited for the analysis done here. The experiments were easy to follow and the analysis was clear.

**Interesting findings**. The main findings related to underestimation, comparison of training-set and test-set dynamics during training, and dependence on training data were interesting and not obvious. The finding that perturbed, unnatural sequence received unusually high probabilities (while test samples receive unusually low probabilities) [Figure 4] was especially interesting.

**Clarity**. The paper was well-written and enjoyable to read. The authors articulated how their work fits in with related work, and the concepts (e.g. the LNRE zone), methodology, and results are clearly written. Well done.

**Additional experiments**: A good addition would be measuring the impact of model size with models larger than the target distribution. What would GPT2-XL's or GPT-3's probability assignments look like on a target distribution from GPT2-medium?

**Summary Of The Paper:**

This paper investigates how language models allocate their probability mass, with an emphasis on rare sequences that are part of the 'heavy tail' of the distribution of natural language sequences. The authors use a language model to define a target distribution with access to samples and ground-truth probabilities. A language model is trained on an empirical estimate of the target distribution, and the authors study the gap between the learned model's and target distribution's probability assignments.

Using this methodology, the authors uncover various interesting phenomena: the model systematically assigns lower probabilities than the target distribution, but assigns unusually high probabilities to unnatural, perturbed sequences (suggesting an explanation for where the probability mass moved to). The authors include several fine-grained analyses with additional interesting findings.

**Summary Of The Review:**

This investigatory paper defines the problem that they are studying, develops a methodology for studying it, and clearly analyzes the results. The investigation yields interesting findings related to language models. This paper would make a great addition to ICLR - accept.

---

> ### Author Response · Authors · 2021-11-15
> **Response to Reviewer V22X**
>
> Thank you for the kind and thoughtful comments. Please see our comment below.
>
> **Additional experiments with models larger than the target distribution**: Thank you for this suggestion. We have deployed training runs on larger models and will include these results in the final version of the paper.

---

### Author Response · Authors · 2021-11-19
**Further Questions from Reviewers?**

Dear Reviewers,

We just wanted to ask if you had any further questions for us. We’d be happy to address any lingering concerns. Thanks!

---

### Decision · Program_Chairs · 2022-01-20

**Decision:**

Accept (Poster)

**Comment:**

This paper presents a very interesting study of using an artificial language (generated using a specific algorithm via a transformer model) and training SOTA transformer and LSTM language models on that language;  the authors show that these LMs underestimate the probability of sequences from this language and overestimate the probability of ill formed sentences, among other observations.  This is a very interesting study that captures the behavior of recent LMs.  All reviewers are supportive of accepting this paper and it is good to see the engagement between reviewers and authors of this paper.